# Canola Meal as Raw Material for the Development of Bio-Adhesive for Medium Density Fiberboards (MDFs) and Particleboards Production

**DOI:** 10.3390/polym14173554

**Published:** 2022-08-29

**Authors:** Jean Lawrence Tene Tayo, Robert Jakob Bettelhäuser, Markus Euring

**Affiliations:** Department of Forest Botany and Tree Physiology, Georg-August University Goettingen, 37073 Goettingen, Germany

**Keywords:** canola meal, protein, bio-adhesive, wood-based panels, renewable resources, MDF, particleboard

## Abstract

In the context of natural resource scarcity, environmental challenges and human health concerns, the development of alternative solutions becomes crucial to sustainable development. Sustainable and renewable protein-containing materials such as soy or canola have proved to have wood bonding properties comparable to those of synthetic binders. In addition, the availability of canola meal offers a great possibility for the development of bio-adhesives for the wood-based panel industry. Furthermore, direct utilization of canola meal helps to avoid expensive and low-yield protein isolation processes. Using three different solvent solutions (water and 1 mol and 2 mol sodium hydroxide), canola-based bio-adhesives were prepared and used for the production of medium-density fiberboards (with 10 mm thickness and 800 kg/m^3^ target density) and three-layer particleboards (with 15 mm thickness and 640 kg/m^3^ target density). The produced boards were tested for their mechanical properties and dimensional stability according to European norms. With the MDFs’ bending strength values above 40 N/mm^2^ and internal bonding strength greater than 0.5 N/mm^2^, the results show that there is indeed a possibility to achieve good mechanical properties using canola meal as a binder. The use of NaOH solutions as denaturants, as well as the addition of colasol, helped improve the bonding properties of the boards by 35.49% and 64.52% for 1 mol and 2 mol NaOH solutions, respectively. The obtained results show that the developed canola-based bio-adhesive can compete with conventional ones. However, despite the good mechanical properties of the produced boards, their poor dimensional stability due to the low water resistance of natural proteins suggests further improvement for industrial application.

## 1. Introduction

Recent growing concerns about public health, coupled with environmental issues, are constantly pushing the wood-based panel industry to restructure itself in order to adapt to new regulations while maintaining the product’s standards and performance. The need to replace conventional petroleum-based adhesives, which have proved to have a significant impact on the environment as well as on human health, is now leading to the redevelopment of bio-based adhesives. These adhesives are produced from natural resources that are not only renewable but also health- and environment-friendly. The use of bio-adhesives goes back to the beginning of human technological development [1]. Unfortunately, the setbacks (such as low durability and low water resistance) of these binders led to the development of synthetic adhesives based on gas and petroleum residues in the late 1950s. Due to the higher performance, fast curing rate and low cost of synthetic adhesives, bio-based binders became less competitive and were no longer in use. However, these synthetic adhesives have proved to represent a danger to human health since they emit formaldehyde, a substance classified by the International Agency for Research on Cancer as a ‘known human carcinogen’. Therefore, legislations concerning both work environment and final product emissions have steadily become stricter over time [2]. Moreover, the fact that these adhesives are based on petroleum residues makes them neither environment-friendly nor sustainable. In the last three decades, there has been a great need for research on bio-based adhesives, essentially those based on agricultural and forest resources and by-products [3]. Many scholars have investigated the use of renewable resources for the development of bio-based adhesives. Among these resources are tannin (hydrolyzable or condensed tannin), lignin (kraft lignin, organosolv and lignosulfonates), wood (fibers, extractives and bio-oil), starch (corn, cassava and plant-based), plants (proteins and polymers), soy (soy flour, soybean and soy protein) and albumin. As for oilseeds (such as soybean or canola meal), they have so far received little interest despite their great potential to be used as raw material for the development of bio-based adhesives. In this category, soybean flour or protein has been the most experimentally studied oilseed-based raw material for bio-based adhesives. From 275 studies selected by Arias et al. [4] for a peer review on bio-based adhesive, 66 focused on soy as raw material. Surprisingly, none of the 275 publications mentioned the use of canola despite its high availability and its potential suitability for the development of bio-adhesives for the wood-based industry.

According to Statista [5], canola was the second most produced type of oilseed in 2021 (70.62 million metric tons) after soybeans (363.86 million metric tons). Canola is mainly cultivated for the production of canola oil, which is used either in the food market (cooking oil) or in the production of biodiesel for the fuel market. Residues of the oil production, also called canola meal, are mostly used in the livestock industry as feed or in agriculture as fertilizer for soil amendment. Depending on the breed and the growing conditions, canola seeds contain 37.5% to 50.9% oil [6]. This means that 34.67 to 44.14 million metric tons of canola meal was available in 2021, making it one of the most available raw materials for bio-adhesives. Moreover, because of the presence of glucosinolates, erucic acid, phytates and phenolics, canola meal is unsuitable for human consumption and therefore can expand capabilities for the bio-renewable economy and lessen dependence on fossil fuel feedstocks [6]. Canola meal contains about 30–50% protein, which consists of 50% cruciferin, 20–40% napin and up to 8% oleosin [7]. Similar to soy protein, canola protein has abundant reactive groups, such as carboxyl, hydroxyl, amino, disulfide, amidazole, indole, phenolic and sulfhydryl groups, on the molecule chain [8], which make it suitable for bio-adhesive development.

Though canola might have great potential as a substitute for conventional adhesives, studies on its suitability for the development of bio-adhesives remain scarce. So far, only a handful of experiments have been carried out with protein isolates [6,9,10,11,12], canola flour [13,14,15,16,17,18] or canola oil [19], either alone or in combination with synthetic binders. Canola protein-based adhesives have so far shown better strength properties compared to crude meal-based ones. However, the protein isolation process is a costly one. Using canola meal helps to avoid expensive and low-yield protein isolation processes [20]. The present study thus focuses on the suitability of canola meal as raw material for the development of bio-adhesive for MDF and particleboard production.

## 2. Materials and Methods

### 2.1. Material

The canola meal used to conduct this experiment was obtained from Kleeschulte GmbH & Co. KG (Büren, Germany). This by-product of the canola oil manufacturing process arrived in the form of pellets. Colasol, a type of animal gluten, was provided by Fritz Häcker GmbH + Co. KG (Vaihingen/Enz, Germany). This product has a solid content of 50%. The glycerin was also provided by Fritz Häcker GmbH + Co. KG. The hydrophobic agent used for the fiberboards was HydroWax 138 for MDF and HydroWax 138 for particleboards. They were purchased from SASOL Wax GmBH (Hamburg, Germany) and had a solid content of 50%. The urea was purchased from SKW Stickstoffwerke Piesterizt Gmbh, Lutherstadt Wittenberg, Germany. The NaOH was purchased from VWR Chemicals, Darmstadt, Germany. The wood fibers used for this experiment were of TMP type obtained from GUTEX Holzfaserplattenwerk H. Henselmann GmbH & Co. KG (Waldshut-Tiengen, Germany). This product is a mixture of 80% Norway spruce (*Picea abies*) and 20% silver fir (*Abies alba*). The wood particles were provided by Pfleiderer GmbH (Ingolstadt, Germany).

### 2.2. Methods

#### 2.2.1. Preparation of Canola Flour-Based Adhesives

The first step consisted of crushing the canola pellets and sieving them through a 400 µm mesh. The canola meal obtained had a moisture content of 9.8%. Chemical denaturation of protein using an alkaline solution has proven to be very an efficient way to modify the structure of the protein [10,21], and canola protein is highly soluble in an alkaline environment [10]. Two alkaline solutions of 1 mol and 2 mol NaOH concentrations were prepared and used to chemically modify the structure of the canola protein. Water was used as the third solvent in the preparation of the adhesives following the same procedure. Other components such as glycerine, colasol and urea were used in the formulations to enhance the gluing properties of the adhesives. The preparation of the adhesive consisted of thoroughly mixing the canola meal with the corresponding amount of solvent for about three to four minutes to assure a certain homogeneity and a certain degree of protein denaturation. While continuing to stir, the other components (glycerin, colasol and urea, according to the recipe) were added. For urea-containing adhesives, urea granules were first added to the solvent and stirred until completely dissolved before the canola flour and other ingredients were added to the mixture. In order to improve the dimensional stability of the board, hydrowax was added to the slurry shortly before spraying (1% for MDF and 2% for particleboards based on the absolute dry wood material). In total, 24 different adhesive types were made and tested on MDFs, with solid content between 20 and 40%. Table 1 shows the mixing ratios for each adhesive type.

As for the particleboards, only 3 adhesive types were used. These were those that gave the most promising results with the MDFs. The mixing ratios were also a bit altered, with 50% of the solid content consisting of canola powder and the other 50% consisting of either glycerine, colasol or both glycerine and colasol (25% each in this case).

#### 2.2.2. MDF and Particleboard Production and Testing

The lab scale production of MDFs and Particleboards was made as shown on flowchart below (Figure 1). The wood fiber material was dried using a flash tube drying system to a moisture content of about 5% prior to the production of the MDF boards. The particles were oven-dried to 2–3% moisture. The MDF and three-layer particleboards were produced in the pilot plant of the Biotechnikum, Georg-August University of Göttingen. First, the wood fiber material was dried using a flash tube drying system to a moisture content of about 5% prior to the production of the MDF boards. The particles were oven-dried to 2–3% moisture content before being used. The necessary wood fiber quantity was weighed to obtain a final target density of 800 kg/m^3^ and fed into a rotary blending drum. The canola-based adhesives were applied to the fibers using an atomizing nozzle with a needle size of 2.3 mm (model A11 from Krautzberger Company, Eltville am Rhein, Germany). The resin load was 15% based on the oven-dry mass of the wood material. Afterwards, the blended fibers were placed in the flash tube dryer at 60 °C to unlock the fibers and reduce the moisture content. Rectangular mats were hand-formed and manually pre-pressed. The hot-pressing took place in a computer-controlled laboratory-scale hydraulic single-opening hot-press (Siempelkamp Hydraulic Lab Press A 308/1988) with 200 bar pressure at 200 °C (Table 2). After conditioning at room temperature for 24 h, the boards were next trimmed to avoid edge effects and sanded on both sides by using a wide-belt sanding machine (Felder type FW 950 C from Felder Group, Hall In Tirol, Austria) before being tested.

The mechanical properties of the boards were determined according to the European standards EN 310, 1993 and EN 319, 1993 for bending strength (BS) and internal bonding (IB). The tests were performed using a universal testing machine from ZWICK/ROELL (type 10). Eight test samples of 50 mm × 50 mm × thickness were chosen to perform these tests. For thickness swelling (TS) and water absorption (WA), 8 test samples from each of the produced boards were immersed in a clean water bath. After 24 h of immersion, their new weights and thicknesses were measured again. The water absorption and the thickness swelling were subsequently calculated and expressed as a percentage of the original mass or thickness.

Apart from assessing the suitability of canola meal as raw material for environment-friendly binders, this experiment was also designed to examine the effects of the use of different solvents (water, 1 mol NaOH and 2 mol NaOH solutions), as well as the effect of the presence of other ingredients (urea, glycerine and colasol). For this purpose, a full-factorial experimental design (2 × 2 × 2 × 4) was used to evaluate not only the effect of each component but also the interaction effect between them. The statistical analysis of the data was conducted using the RStudio programming package.

## 3. Results and Discussion

### 3.1. Medium-Density Fiberboards

Figure 2 below demonstrates an overview of the density distribution of the internal bonding (IB), bending strength (BS), thickness swelling (TS) and water absorption (WA) values. A great disparity was observed among the values obtained, with IB ranging from 0.06 N/mm^2^ to 1.06 N/mm^2^, BS ranging from 16.9 N/mm^2^ to 74.4 N/mm^2^ and TS ranging from 10.8% to 62.25%. The IB density distribution shows that most of the MDFs could not meet the EU norm EN 622-5, which is 0.6 N/mm^2^. Only 16.38% of the total boards produced could meet the EU requirements for MDF. The BS properties of the MDFs were much more promising, with 97.13% of tested MDFs meeting the European standard (22 N/mm^2^). Moreover, more than 53.45% of the MDF boards had a BS value over 40 N/mm^2^. This suggests that these bio-adhesives can easily be used to produce load-bearing MDFs. However, depending on the use of the end-product and the manufacturing technology, IB might receive minor consideration, as it is not always the most important characteristic that manufacturers focus on [22]. The TS results show that the bonding strength and the interconnection between the fibers are subject to hydrolysis. Only 14.96% of the boards could meet the TS norm (15%).

The results for IB, BS, TS and WA are presented in Table 3. It can be seen from the table that the solvent had a significant effect on the boards’ properties (*p* < 0.05). The use of water as the solvent led to the lowest IB values recorded (0.31 N/mm^2^). Denaturation of the canola meal using alkali solution helped to increase this value by 35.49% and 64.52% for 1 mol and 20 mol solutions (with mean values of 0.42 N/mm^2^ and 0.51 N/mm^2^, respectively). The chemical denaturation of the canola meal also significantly (*p* < 0.001) affected the BS. However, the use of a 2 mol NaOH solution led to a decrease in BS compared to 1 mol, meaning that high denaturation might jeopardize the BS properties. In a similar study with different NaOH and HSO4 concentrations, Yang et al. [17] proved that the decomposition of the chemical structure of canola meal components resulted in the loss of the adhesive properties of the protein.

According to Kuehler and Stine [23], extreme hydrolysis breaks molecules into small molecules, leading to the loss of their adhesive properties. Slight chemical denaturation of the RS protein will thus enhance the adhesive properties of protein-based adhesives, while more pronounced modification will negatively affect them. TS and WA values followed the same trend, with lower mean values obtained with water (18.63 ± 4.05% and 36.89 ± 6.34%, respectively). The TS values increased by 10.6% with the use of 1 mol and 44.4% with 2 mol of NaOH (values increased from 18.63% to 20.55 ± 5.14% and 26.80 ± 7.54%). These results correlate with the findings of Yang et al. [17], who used different denaturant concentrations to develop a canola flour/PF adhesive. A higher level of protein denaturation led to higher vulnerability to hydrolysis.

Contrary to expectations, the use of glycerine in the preparation of the adhesive did not affect the overall mean values of the main adhesive properties (Figure 3). No significant effect was found on either IB (mean value of 0.42 N/mm^2^ for glycerine-containing variants and 0.40 N/mm^2^ for variants without glycerine) or the BS performance (41.3 N/mm^2^ and 40.61 N/mm^2^). In contrast, the use of glycerine enhanced the hydrophobicity of the boards but reduced TS by 8.45% and WA by 10.04%. However, when looking at the interaction effect between glycerine and the solvent, it is evident that the interaction between alkali and glycerine led to an increase in IB of about 10% for both 1 mol and 2 mol groups (values increased from 0.40 N/mm^2^ to 0.44 N/mm^2^ and from 0.48 to 0.53 N/mm^2^, respectively), while the opposite result was observed with water, in which IB decreased by approximately 12.12% with the use of glycerine. As for the BS, the presence of glycerine had a positive but weaker effect with increasing NaOH concentration. The 2 mol group showed a positive reaction to glycerine, while the 1 mol and water groups were negatively affected (TS dropped from 18.86 ± 2.98% to 18.30 ± 4.72%)

The addition of colasol to the slurry had an incredible effect on the adhesive properties, enhancing the bonding and the hydrophobicity at the same time. The presence of this component significantly improved the IB, BS, TS and WA of the produced MDF (*p* < 0.001). An increase in IB of up to 67.74% could be achieved with colasol (0.31 N/mm^2^ for variants without colasol and 0.52 N/mm^2^ for colasol-containing variants). As for the BS, the application of this ingredient also helped improve the MDF strength. Adhesive variants containing colasol had a BS 1.52 times higher than that of variants without colasol. In the presence of this component, the IB increased by 38.46%, 62.5% and 91.43% for water, 1 mol and 2 mol solvent solutions, respectively, suggesting that the alkali concentration also favors the curing performance of the colasol-containing adhesives. Similarly, the presence of colasol helped to improve the load-bearing capacity (BS) of the MDF, with an increase in alkali concentration yielding the best results. An increase in BS of 36.53% with water, 40.44% with 1 mol and 74.29% with 2 mol NaOH solution was made possible with the combination of colasol.

TS was also impacted by the presence of colasol, with the greatest effect registered with 1 mol solvent solution. An improvement in TS of 9.24%, 30.97% and 27.98% for water, 1 mol and 2 mol solutions, respectively, was achieved when using colasol. However, it should be noted that, although this improvement was less important with water as the solvent solution, adhesives made with water had the best water resistance properties, with a mean value of 18.58% compared to 20.54% and 26.8% for 1 mol and 2 mol alkali solutions, respectively. This result suggests that there is a negative correlation between protein denaturation and the water resistance of the protein–protein or protein–wood bonds.

In contrast, urea had a negative effect on the boards’ properties, probably because of the alkaline nature of urea, which surely contributed to the extreme denaturation of the canola protein, affecting the bonding properties, as well as the hydrophobicity of the boards. A decrease of 6.98% (from 0.43 N/mm^2^ to 0.40 N/mm^2^) in IB was observed when using urea. The BS was also reduced by 16.2% in the presence of urea. TS was also negatively affected by urea, with an increase of 15.65% (from 20.38% to 23.57%).

The interaction between glycerine and colasol had an overall positive effect on the MDF properties. In the presence of both elements, the mean values of IB, BS and TS were 0.49 N/mm^2^, 47.98 N/mm^2^ and 19.82%, respectively. In their absence, these values dropped to 0.27 N/mm^2^ for IB and 32.52 N/mm^2^ for BS, while TS increased to 27.77%. However, the results show that the presence of glycerine was a drawback for the effect of colasol. In fact, better performance was achieved when using colasol without glycerine, with 0.54 N/mm^2^ for IB, 50.08 N/mm^2^ for BS and 18.11% for TS. The combined effect of glycerine and urea showed the same trend, with urea having a drawback effect. The same observation was made on the interaction effect between colasol and urea, with the latter still having a negative effect.

### 3.2. Particleboards

As for the particleboards, although the level of protein hydrolysis had no significant effect on the IB values, a higher bonding strength was obtained with the 2 mol NaOH solvent solution. While the overall mean IB for 1 mol solvent was 0.27 N/mm^2^, that of 2 mol solvent was 0.29 N/mm^2^. This demonstrates the fact that the more the protein structure is modified, the more the bonding capacity increases due to the availability of more functional groups, which interact with the wood material surface. This effect becomes significant when taking into consideration the interaction effect of either glycerine or colasol. Using colasol, the IB increased to 0.35 N/mm^2^ and 0.43 N/mm^2^ for 1 mol and 2 mol solvents, respectively (Figure 4). Although the interconnection of particles in the particleboard is weaker than that of fibers in MDF due to the particle size and the higher bulk density, the results show that there is a possibility of achieving good bonding strength using modified canola meal adhesive. Unlike glycerine, which seems to act as a lubricant, colasol seems to have a crosslinking effect, which enhances the bonding properties of the protein and helps achieve greater values than the EN requirement (0.35 N/mm2). Being a type of animal gluten, colasol might have the same properties as wheat gluten, which is known to have high amounts of hydrophobic amino acids [24], which help to improve the gluing capacity of the adhesive and reduces the water vulnerability of the protein-wood bonds at the same time.

Though none of the three variants could meet the EN requirement for the load-bearing capacity (11 N/mm^2^), it can be seen that the colasol-containing variant performed much better. As observed with the MDFs, the BS values of the particleboards decreased with increasing NaOH concentration. The canola-bonded particleboards exhibited poor water resistance, with TS values far beyond the EN requirement, which is 15%. As already known, the stability of hydrogen bonds in bio-adhesives is extremely vulnerable to water, which weakens or breaks down the intermolecular forces [25]. Moreover, the fact that the majority of the amino acids present at the surface of canola protein molecules are hydrophilic [26] enhances the water vulnerability of the adhesive. Although a 2% hydrophobic agent was applied, poor stability of the protein–wood connection was observed. This poor water resistance of the particleboards is explained by the low IB strength. As observed with the MDF, there is a negative correlation between the IB strength and TS. The higher the IB, the lower the TS. This means that achieving a higher IB will surely help reduce the water vulnerability of the protein-bonded board.

The overall poor performance of the particleboards, especially the very low bending strength, might be caused by the oven-drying process after particle blending. Because of the high moisture content of the blended wood particles, they had to be dried for 2–2.5 h to bring the moisture content to 9–11%. Although this effect was not assessed, it is suspected that the long drying time might have affected the bonding capacity of the adhesive.

## 4. Conclusions

The aim of the present study was to assess the suitability of canola meal for the development of bio-adhesives for the production of MDF and particleboards using a simple protein modification process. Using ≤400 µm fractions of canola meal provided good results for MDFs, which achieved over twice the EN norm for the load-bearing capacity (mean value above 0.4 N/mm^2^). The IB strength of MDFs could be increased with a high concentration of NaOH and the addition of colasol. Indeed, the use of 2 mol NaOH as a solvent and colasol helped achieve IB values above 0.8 N/mm^2^, which is greater than the EN requirement (0.6 N/mm^2^). Unfortunately, a higher level of hydrolysis negatively impacted the water resistance of the MDFs, leading to poor water resistance of the boards (TS values above 15%). Low IB and BS values, as well as high water vulnerability of the particleboard, might be due to the short pressing time. Nevertheless, colasol-containing variants exhibited promising IB strength (0.43 N/mm^2^). Despite the fact that some of the mechanical and dimensional stability properties of both MDFs and particleboards met EN standards, there is room for improvement. Therefore, further studies are needed to improve the water resistance of the developed binder, as well as its bonding properties. Further investigations should thus be conducted using proper and adequate modification processes or natural crosslinkers that will increase both the bonding properties and the hydrophobicity of the adhesive.

## Figures and Tables

**Figure 1 polymers-14-03554-f001:**
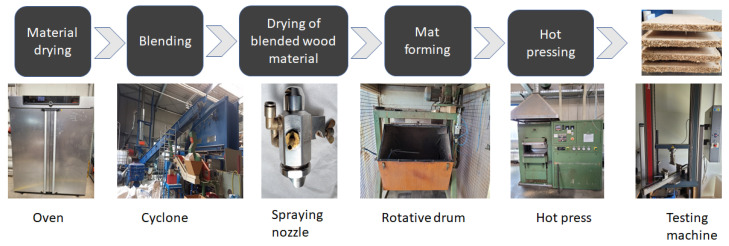
Flowchart for experimental procedure and pictures of some equipment used.

**Figure 2 polymers-14-03554-f002:**
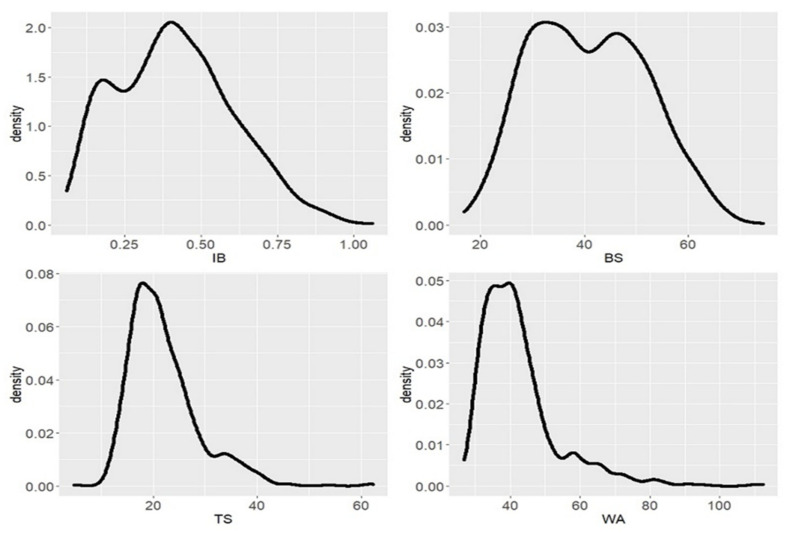
Density distribution of IB, BS, TS and WA values.

**Figure 3 polymers-14-03554-f003:**
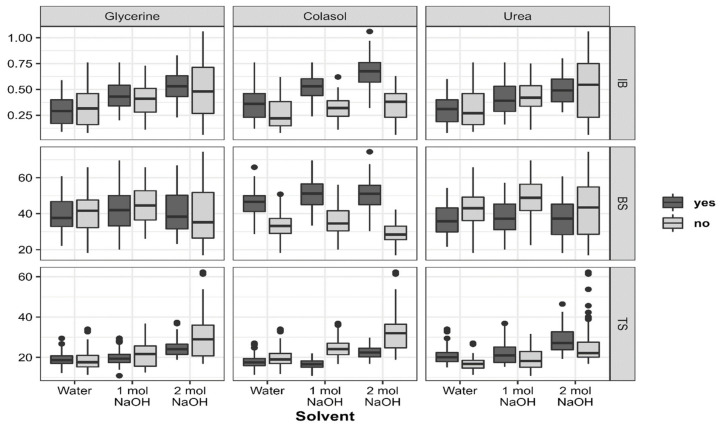
Effects between the solvent, glycerine, colasol and urea on the mechanical properties of the produced MDFs.

**Figure 4 polymers-14-03554-f004:**
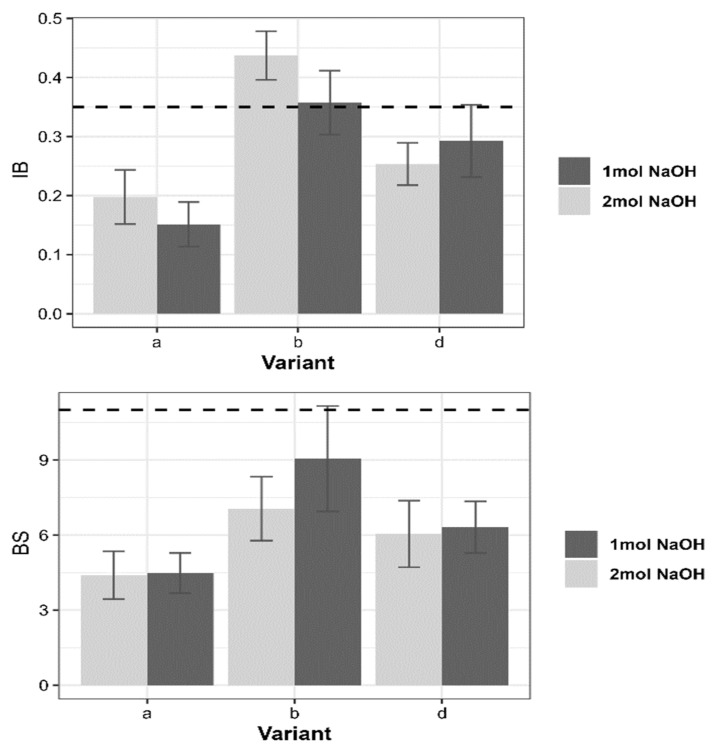
IB and BS of particleboards bonded with canola meal-based adhesive.

**Table 1 polymers-14-03554-t001:** Mixing ratios for each of the adhesive formulations.

Variant	Canola Meal (%)	Glycerine (%)	Colasol (%)	Urea (%)
0	100	-	-	-
A	44	56	-	-
B	44	-	56	-
C	44	-	-	56
D	35	32.56	32.56	-
E	35	-	32.56	32.56
F	35	32.56	-	32.56
G	36	21.6	21.6	21.6

Note: (1) The numbers 0, 1 and 2 are used to differentiate solvents (e.g., 0 means water as solvent, 1 means 1 mol NaOH solution as solvent and 2 means 2 mol NaOH solution as solvent). (2) The values in the table account for the solid content.

**Table 2 polymers-14-03554-t002:** MDF and particleboard production parameters.

Parameters	Board Type	
	MDF	Particleboard
Board’s target density	800 kg/m^3^	640 kg/m^3^
Board thickness	10 mm	15 mm
Resin load	15%	15%
Hydrowax	1%	2%
Pressing pressure	200 bars	200 bars
Pressing temperature	200 °C	200 °C
Pressing time	6 min	3.5 min
Mat moisture before pressing	16–18%	9–11%
Board per variant	4	4

For the particleboards, the blended particles were oven-dried at 60 °C to a moisture content of 9 to 11%. Mat forming and hot-pressing were performed in the same way as for the MDF.

**Table 3 polymers-14-03554-t003:** Overview of the mechanical and dimensional properties of MDFs bonded with canola meal-based adhesives.

Solvent	Variant	IB (N/mm^3^)	BS (N/mm^3^)	TS (%)	WA (%)
Water	0	0.15 ± 0.04	38.13 ± 8.20	14.78 ± 1.80	33.35 ± 7.35
	a	0.28 ± 0.14	35.20 ± 4.21	18.83 ± 2.12	36.09 ± 2.68
	b	0.41 ± 0.22	49.47± 4.70	16.19 ± 4.25	35.55 ± 6.18
	c	0.22 ± 0.10	29.36 ± 4.04	23.69 ± 3.96	44.96 ± 6.89
	d	0.30 ± 0.13	48.55 ± 5.29	16.61 ± 2.24	33.55 ± 2.53
	e	0.41 ± 0.07	43.99 ± 5.24	18.55± 2.61	36.27 ± 4.88
	f	0.24 ± 0.11	31.57 ± 4.88	20.62± 2.35	36.61 ± 3.65
	g	0.31 ± 0.08	41.27 ± 6.94	19.40 ± 2.63	37.10 ± 5.38
1 mol NaOH	01	0.27 ± 0.15	42.45 ± 5.80	25.87 ± 2.45	43.73 ± 2.45
	a1	0.35 ± 0.08	38.84 ± 8.06	19.35 ± 1.93	37.03 ± 3.41
	b1	0.52 ± 0.12	57.06 ± 5.01	14.43 ± 1.08	32.08 ± 2.30
	c1	0.26 ± 0.07	32.99 ± 3.67	27.97 ± 4.55	48.80 ± 7.58
	d1	0.53 ± 0.10	55.25 ± 6.11	17.06 ± 2.46	34.58 ± 4.10
	e1	0.49 ± 0.10	45.97 ± 5.64	17.28 ± 4.00	35.58 ± 4.07
	f1	0.35 ± 0.07	29.73 ± 4.28	24.08 ± 2.73	45.37 ± 4.50
	g1	0.53 ± 0.12	43.99 ± 5.38	18.39 ± 1.54	37.27 ± 3.96
2 mol NaOH	02	0.32 ± 0.12	25.32 ± 4.40	38.07 ± 7.90	73.47 ± 13.84
	a2	0.44 ± 0.09	32.71 ± 4.58	21.69 ± 1.86	43.54 ± 3.86
	b2	0.81 ± 0.10	57.66 ± 6.22	18.89 ± 1.64	40.14 ± 2.54
	c2	0.38 ± 0.06	26.90 ± 3.84	36.29 ± 3.64	66.02 ± 8.62
	d2	0.68 ± 0.08	54.19 ± 5.56	22.90 ± 2.33	42.56 ± 4.05
	e2	0.59 ± 0.10	46.32 ± 4.68	23.37 ± 3.43	45.45 ± 5.93
	f2	0.43 ± 0.08	31.42 ± 4.65	28.61 ± 3.53	52.38 ± 6.78
	g2	0.59 ± 0.09	44.65 ± 6.72	24.61 ± 2.74	45.63 ± 4.43

## Data Availability

The data presented in this research are available upon request from the corresponding author.

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
