# Peer review of "Canola Meal as Raw Material for the Development of Bio-Adhesive for Medium Density Fiberboards (MDFs) and Particleboards Production"

_polymers, 2022, doi:10.3390/polym14173554_

Round 1

Reviewer 1 Report

1. Provide the details of Particleboards production process with the aid of a diagram.

2. Also, include the images of the machines used. 

3. Include numerical results in the conclusion section.

Author Response

Dear Reviewer,

thank you very much for supporting our recent research study!

Please find attached our reponse to your comments.

Best wishes,

the authors

Reviewer 2 Report

Overall the paper was written in good manner. However, there are room for improvements should be considered by the authors in the revised manuscript.

1. The abstract is well written. However, no discussion on specific and overall findings in this section. Additionally, a brief conclusion has to added in the abstract.

2. The introduction should be include research motivations, gaps and problem statements. It seems the paper is lacked on this part. Please improve.

3. Enlarge and improve the image resolution of Figure 1 to make clearer.

4. Conclusion should be include with future outlook and future research. Please add this part.

Author Response

(The authors gave the same response as above.)
